# Antioxidant and Cytotoxic Properties of *Berberis vulgaris* (L.) Stem Bark Dry Extract

**DOI:** 10.3390/molecules29092053

**Published:** 2024-04-29

**Authors:** Ionuț Mădălin Ivan, Octavian Tudorel Olaru, Violeta Popovici, Carmen Lidia Chițescu, Liliana Popescu, Emanuela Alice Luță, Elena Iuliana Ilie, Lorelei Irina Brașoveanu, Camelia Mia Hotnog, George Mihai Nițulescu, Rica Boscencu, Cerasela Elena Gîrd

**Affiliations:** 1Faculty of Pharmacy, University of Medicine and Pharmacy “Carol Davila”, Traian Vuia 6, 020956 Bucharest, Romania; ionut.ivan@drd.umfcd.ro (I.M.I.); liliana.popescu22@umfcd.ro (L.P.); emanuela.luta@umfcd.ro (E.A.L.); elena.ionita@drd.umfcd.ro (E.I.I.); george.nitulescu@umfcd.ro (G.M.N.); rica.boscencu@umfcd.ro (R.B.); cerasela.gird@umfcd.ro (C.E.G.); 2Center for Mountain Economics, “Costin C. Kiriţescu” National Institute of Economic Research (INCE-CEMONT), Romanian Academy, 725700 Vatra-Dornei, Romania; 3Faculty of Medicine and Pharmacy, “Dunărea de Jos” University of Galați, A.I. Cuza 35, 800010 Galați, Romania; carmen.chitescu@ugal.ro; 4Center of Immunology, “Stefan S. Nicolau” Institute of Virology, Romanian Academy, 285 Mihai Bravu Ave., 030304 Bucharest, Romania; lorelei.brasoveanu@virology.ro (L.I.B.); camelia.hotnog@virology.ro (C.M.H.)

**Keywords:** *Berberis vulgaris* (L.) stem bark, dry hydro-ethanolic extract, phenolic secondary metabolites, berberine, antioxidant activity, cytotoxicity

## Abstract

*Berberis vulgaris* (L.) has remarkable ethnopharmacological properties and is widely used in traditional medicine. The present study investigated *B. vulgaris* stem bark (Berberidis cortex) by extraction with 50% ethanol. The main secondary metabolites were quantified, resulting in a polyphenols content of 17.6780 ± 3.9320 mg Eq tannic acid/100 g extract, phenolic acids amount of 3.3886 ± 0.3481 mg Eq chlorogenic acid/100 g extract and 78.95 µg/g berberine. The dried hydro-ethanolic extract (BVE) was thoroughly analyzed using Ultra-High-Performance Liquid Chromatography coupled with High-Resolution Mass Spectrometry (UHPLC–HRMS/MS) and HPLC, and 40 bioactive phenolic constituents were identified. Then, the antioxidant potential of BVE was evaluated using three methods. Our results could explain the protective effects of Berberidis cortex EC_50_FRAP = 0.1398 mg/mL, IC_50_ABTS = 0.0442 mg/mL, IC_50_DPPH = 0.2610 mg/mL compared to ascorbic acid (IC_50_ = 0.0165 mg/mL). Next, the acute toxicity and teratogenicity of BVE and berberine—berberine sulfate hydrate (BS)—investigated on *Daphnia* sp. revealed significant BS toxicity after 24 h, while BVE revealed considerable toxicity after 48 h and induced embryonic developmental delays. Finally, the anticancer effects of BVE and BS were evaluated in different tumor cell lines after 24 and 48 h of treatments. The MTS assay evidenced dose- and time-dependent antiproliferative activity, which was higher for BS than BVE. The strongest diminution of tumor cell viability was recorded in the breast (MDA-MB-231), colon (LoVo) cancer, and OSCC (PE/CA-PJ49) cell lines after 48 h of exposure (IC_50_ < 100 µg/mL). However, no cytotoxicity was reported in the normal epithelial cells (HUVEC) and hepatocellular carcinoma (HT-29) cell lines. Extensive data analysis supports our results, showing a significant correlation between the BVE concentration, phenolic compounds content, antioxidant activity, exposure time, and the viability rate of various normal cells and cancer cell lines.

## 1. Introduction

Numerous pharmaceutical companies are focused on researching and developing new formulations based on herbal sources, which can help manage chronic diseases. The World Health Organization also supports conventional plant-based treatments due to their accessibility, safety for long-term uses, and relatively low production costs. This shift toward natural remedies occurred mainly because some synthetic pharmaceutical drugs may have harmful side effects when used for the long-term treatment of chronic diseases [1]. Therefore, based on traditional medical systems (Ayurvedic and Chinese), phytotherapy in chronic disorders is currently used as an alternative treatment worldwide. Berberis is a significant plant genus with approximately 500 species worldwide. It belongs to the *Berberidaceae* family and has considerable potential applications in the food and pharmaceutical industries [2]. *Berberis* species are native to central and southern Europe, Asia (including the northern zones of Pakistan and Iran), and the north-eastern area of the United States. *Berberis vulgaris* (L.), known as European barberry, common barberry, or Épine-Vinette, has an essential role in herbal therapy; its different parts (fruits, leaves, roots, stem, branches, stem/root bark) have been used in traditional medicine for more than 2500 years. This species can be helpful in various inflammations, high blood pressure, gastrointestinal diseases, hepatic disorders, and diabetes. Numerous studies show that *B. vulgaris* has valuable pharmacological properties, such as antioxidant, antihyperglycemic, anticholinergic, hypolipidemic, anti-inflammatory, anticancer, and antimicrobial properties. In homeopathy, *B. vulgaris* is mainly used in urinary lithiasis, dermatology, rheumatism, and liver diseases [3]. Berberine, the specific isoquinoline alkaloid mainly extracted from common barberry root and stem barks, is formulated for oral administration alone or in various combinations. The administration of berberine-based phytotherapeutics could have a beneficial impact on lipid and carbohydrate metabolism, particularly on glucose homeostasis, being helpful in weight loss, diabetes mellitus [4] and endocrine disorders, liver diseases [5], cardiovascular diseases, atherosclerosis, neurodegenerative diseases, rheumatic diseases, and infectious diseases [1]. Several studies reported berberine-induced toxicity in humans and mice [1]. However, toxic phenomena could be diminished through berberine combination with other phytochemicals or plant extracts. Synergistic effects would also be expected in adequate combinations [1]. Moreover, *B. vulgaris* and berberine display anticancer effects through various cell signaling pathways’ modulation [6], diminishing tumor cell viability and reducing their multiplication in various neoplasia (lung, breast, ovary, gastric cancer) [1]. 

In the present study, we aimed to investigate the hydro-ethanolic dry extract of *B. vulgaris* stem bark (BVE), obtained using a reflux extraction process in 50% ethanol, rotary evaporation, and freeze-drying. Here, 50% ethanol was used as an extraction solvent for obtaining the dry plant extract because of its effectiveness in extracting a broad range of phytochemicals (polar, moderately polar, and some nonpolar compounds); in addition, it has a low toxicity profile compared to other solvents (methanol, acetone, hexane, ethyl acetate or chloroform). Reflux extraction is a low-cost and efficient tool that ensures a high content of bioactive constituents through the consequent rotary evaporation and freeze-drying. The lyophilization (freeze-drying) process provides substantial stability to plant extracts by preserving the secondary metabolites with antioxidant activity.

A complex analysis of BVE’s phenolic compounds was performed using ultra-high-performance liquid chromatography coupled with high-resolution mass spectrometry (UHPLC–HRMS/MS). Berberine was quantified through HPLC-DAD. The BVE’s antioxidant potential was in vitro evaluated through the free radical scavenging (DPPH and ABTS) and reducing power (FRAP). The acute toxicity of BVE and berberine—berberine sulfate hydrate (BS)—was assessed in vivo on two *Daphnia* species. In contrast, their teratogenic potential was evaluated by applying the embryo test to *Daphnia magna* embryos. Moreover, the antitumor potential of BVE and BS was investigated in several human cancer cell lines: hepatocellular (HEP G2), colon (LoVo and HT-29), breast (MDA-MB-231), ovary (SK-OV-3), and tongue (PE/CA-PJ49), using classical oncolytic drugs as positive controls. Extensive data analyses support our results, showing significant correlations between the BVE concentration, exposure time, phenolic constituent content, antioxidant activity, and cytotoxicity.

## 2. Results and Discussion

### 2.1. Phenolic Compounds (Polyphenols and Phenolic Acids) Quantification

Berberidis cortex dry hydro-ethanolic extract was obtained with a yield of 16.35%. Other studies reported similar yields: 18.7% for roots and 14.7% for leaf extracts in ethanol [6]. Ethanol’s availability and regulatory approval make it an obvious choice due to its balance of effectiveness, safety, and applicability. Moreover, ethanol’s moderate boiling point makes it easy to remove by evaporation, simplifying the process of concentrating the extract and falling within the trend of implementing green technology and using green solvents, which are much safer for the environment. The rotary evaporator could be connected with a vacuum pump, which decreases the boiling point of ethanol (78.2 °C) and facilitates its evaporation. After ethanol collection, it could be subjected to fractional distillation to remove the moisture content and increase its purity.

The standard calibration curves are displayed in Appendix A, while the TPC and TPA values are presented in Table 1. BVE is rich in total polyphenols (TPC = 17.6780 ± 3.9320 mg Eq tannic acid/100 g extract); however, it has shown a phenolic acid content (TPA) of only 3.3886 ± 0.3481 mg Eq chlorogenic acid/100 g extract.

The literature data show that the TPC in various *B. vulgaris* extracts is very different. Our hydro-ethanolic extract of Berberidis cortex has a TPC of 1767.80 mg/g, while El-Zahar et al. [7] reported much lower TPC levels in ethanol extracts of roots (147.2 mg/g) and leaves (120.7 mg/g). Och et al. [8] indicated similar TPC values quantified in 80% methanol extracts of various *B. vulgaris* parts: 58.5 mg/g for the leaf extract, 57.7 mg/g for the stem one, and 52.8 mg/g for the fruit extract. 

### 2.2. Identification and Quantification of BVE Phytoconstituents by UHPLC–HRMS/MS and HPLC-DAD

Table 2 registers all the phytochemicals identified in BVE.

Some constituents were quantified, with gallic acid having the highest amount (540.00 µg/g). It is followed, in decreasing order, by naringenin (90.41 µg/g), berberine (78.95 µg/g), rutin (72.41 µg/g), kaempferol (68.24 µg/g), and galangin (67.21 µg/g). 

Figure 1A displays the chromatogram of the primary phytochemicals identified in BVE by UHPLC-MS, and Figure 1B shows the HPLC-DAD chromatogram of BVE, where berberine has an RT = 32.513.

In Argentinian/Patagonian barberry fruit (*Berberis microphylla*) ethanol extract, Boeri et al. [9] reported the highest amounts of quercetin (1134.54 µg/g), caffeic acid (1092.75 µg/g), and syringic acid (368.55 µg/g). In contrast, gallic acid was 48.17 µg/g. Berberidis cortex is a significant and frequently used crude drug registered in the “Drug Standards of Tibetan Medicines” since 1995. The specific bioactive compounds are alkaloids (berberine, magnoflorine, jatrorrhizine, palmatine), and the most known is berberine, quantified in our BVE (78.95 µg/g). HPLC analysis of methanolic extract of Berberidis cortex from China, harvested from different geographical zones, recorded a range of 21.12–37.5 µg/g berberine [10]. Our results indicated a value twice as high as the first one.

### 2.3. Antioxidant Activity

Table 2 shows significant differences between the IC_50_/EC_50_ values determined by all three methods, IC_50_DPPH = 0.2610 mg/mL, IC_50_ABTS = 0.0442 mg/mL, and EC_50_FRAP = 0.1398 mg/mL, compared to ascorbic acid (IC_50_ = 0.0165 mg/mL). Similar values were reported for *B. microphylla* ethanol extract [9]: ABTS IC_50_ = 0.26 mg/mL and DPPH IC_50_ = 0.38 mg/mL. The substantial antioxidant potential is underlined by the phenolic constituent content and berberine and other alkaloids, which are known for their protective activity [11]

### 2.4. 48-h Acute Toxicity Test Using Daphnia Magna and Daphnia Pulex

After 24 h, *D. magna*’s total lethality was recorded at concentrations ≥ 25 µg/mL BVE and ≥10 µg/mL BS. Similarly, *D. pulex*’s total lethality occurred at concentrations ≥ 50 µg/mL BVE and ≥25 µg/mL BS. Our results revealed that the BS toxicity is higher than BVE for both *Daphnia* species, with *D. magna* being more vulnerable than *D. pulex*. The time-dependent toxicity is more evident in *D. magna* than in *D. pulex.* (Figure 2). The lethality curves’ analysis revealed that, in the *D. magna* bioassay (Figure 2A), BS exhibited a lower LC_50_ value at 24 h compared to BVE, suggesting the higher toxicity of the pure alkaloid, which was expected. However, at 48 h, the BVE toxicity significantly increases, almost to the same potency as BS. After 48 h of exposure, the concentration–response curves for both tested solutions in *D. pulex* displayed similar trends to those recorded in *D. magna* but with differences in the magnitude of lethality (Figure 2B).

### 2.5. Daphnia Magna Embryonic Development Assay

Following the acute toxicity test results, the embryo assay was performed at non-lethal concentrations (2.5 µg/mL BS and 3.125 µg/mL BVE). Minor differences were observed after 24 h (Figure 3a,b). After 48 h, BS stimulated the development of all the embryos (Figure 3c), while BVE exhibited a significant inhibitory effect (Figure 3d), which could be due to the extract’s complex composition. Only 20% of the embryos treated with BVE were fully developed, compared to 90% recorded for those exposed to BS. The mobility and viability of neonates developed in BS solution were similar to those of the control. However, they all failed to form the compound eye, even after 48 h of exposure, suggesting a potential developmental risk. 

In *Daphnia magna*, Vesela et al. [12] reported that the berberine chloride toxicity recorded an LC_50_ of 0.903 µg/mL after 24 h and 0.822 µg/mL at 48 h exposure. Our berberine sulfate hydrate recorded 9.7 µg/mL and 5.3 µg/mL, respectively. In another study on another crustacean [13], 7 µg/mL berberine chloride induced 100% lethality in *Artemia salina* larvae. These differences could be explained by berberine salt, the animal model species, and the provenance. The *D. magna* embryos failed to form compound eyes after BS and BVS treatment. Natural berberine also affects cardiovascular system morphogenesis and functionality in Zebrafish embryos [14]. Based on these findings, in the *Medicinal Plants Monograph Volume 4* [15], the WHO mentions the potential side effects of berberine on humans after consuming more than 500 mg.

### 2.6. In Vitro Anticancer Activity

The antiproliferative activity induced by BVE was evaluated in vitro through several cytotoxic assays by applying different BVE and BS concentrations (6.25–400 µg/mL) to cells derived from six tumor cell lines of different histological origin: HEP G2, LoVo, HT-29, MDA-MB-231, SK-OV-3, and PE/CA-PJ49. Human umbilical vein endothelial cells (HUVECs) were selected as the reference normal cells. 

The BVE and BS antiproliferative capacities, as tested on normal human cells and tumor cell lines, are shown in Table 3. 

The IC_50_ values displayed in Table 3 could be interpreted according to Hidayat et al. [16], resulting in an overview of the BVE and BS cytotoxicity in various cell lines. In normal endothelial cells (HUVEC), they have no cytotoxicity after 24 and 48 h (IC_50_ >> 400 µg/mL). The same interpretation is also available for HT-29 tumor cells, which showed no significant decrease in viability after the BVE/BS treatments.

Generally, BVE exhibited moderate cytotoxicity in the other tumor cells. The most substantial effect, with the lowest IC_50_ values after 24 and 48 h (IC_50_ > 100 µg/mL, respectively, IC_50_ > 50 µg/mL) was seen in breast cancer cells (MDA-MB-231) and OSCC ones (PE/CA-PJ49). In LoVo cells (colon cancer), the cytotoxicity at 24 h was appreciably lower (IC_50_ > 200 µg/mL) but after 48 h of exposure was moderate, similar to the previous ones (IC_50_ > 50 µg/mL). BVE exhibited the lowest effect on human ovary cancer (SK-OV-3 cells) and hepatocellular carcinoma (HEP G2 cells) after 24 and 48 h (IC_50_ > 400 μg/mL and, respectively, >100μg/mL).

Globally, the antiproliferative effects of BS are more potent than those of BVE. In MDA-MB-231 cells, moderate to high cytotoxicity was registered (IC_50_ > 25 and, respectively, >12.5 μg/mL after 24 and 48 h, *p* < 0.05). BS showed similar activity on PE/CA-PJ49 and LoVo (IC_50_ > 50 μg/mL for both exposure times). Moreover, it recorded low toxicity after 24 h contact with SK-OV-3 and HEP G2 cells and a moderate one after 48 h (IC_50_ > 200 μg/mL, respectively, >50 μg/mL). 

The results of in vitro studies are detailed and presented in Appendix A. 

After 24 h, BVE cytotoxicity at the selected concentration range (6.25–400 µg/mL) in normal and tumor cell lines showed significant differences (at α < 0.05, *p*-value was established at 0.0024) between PE/CA-PJ49 and HUVEC, and MDA-MB-231 and HUVEC (*p* = 0.001, Appendix A). Substantial differences (*p* < 0.05) were also observed in the case of HUVEC and all the other tumor cells, except the HT-29 ones, and MDA-MB-231 and HUVEC compared to HT-29 cells (Appendix A). 

After 48 h of exposure to BVE, the percentual values of cell vitality significantly differed between the HUVEC and MDA-MB-231 cells (*p* < 0.001). Appreciable differences (*p* < 0.05, Appendix A) were reported between HUVEC and LoVo, PE/CA-PJ49, and SK-OV-3. Moreover, there were notable differences between HT-29 and LoVo, MDA-MB-231 and PE/CA-PJ49, and HEP G2 and MDA-MB-231 (*p* < 0.05).

In the case of BS, significant differences (*p* = 0.0001 and *p* = 0.000) were recorded exclusively between the HUVEC and MDA-MB-231 cell viability in both periods of exposure (Appendix A). After 24 h, remarkable differences were reported between HUVEC and LoVo, PE/CA-PJ49, and SK-OV-3, and HT-29 and MDA-MB-231 and PE/CA-PJ49 (*p* < 0.05, Appendix A). Moreover, after 48 h, there were notable differences between HUVEC and HEP G2, PE/CA-PJ49, LoVo, and SK-OV-3 (*p* < 0.05); the same was true for HT-29 and PE/CA-PJ49 and MDA-MB-231 (Appendix A). 

No statistically significant differences were reported between the BVE and BS cytotoxicity for the same exposure period in the same cell line (Appendix A).

The cytotoxic activity of BVE was compared to that induced by several drugs (5-Fluorouracil, Cisplatin, and Doxorubicin) [17] that are commonly used in oncological treatments and were applied throughout all the experiments as positive controls. The concentration range used for Cisplatin (CisPt) and 5-Fluorouracil (5-FU) was 3.125–200 µM, while for Doxorubicin (DOX) it was between 0.625 and 40 µM [17], as illustrated in Figure 4 and Figure 5.

The effects of BVE and BS compared to anticancer drugs on normal endothelial cells’ (HUVEC) viability after 24 and 48 h of exposure are displayed in Figure 4. 

The highest HUVEC cell viability diminution was recorded after 24 h at 200 µM CisPt (85.58%), while, at the corresponding concentrations, BVE, BS, and both other drugs did not significantly affect it. At 48 h, 200 µM CisPt reduced the normal cell viability to 55.80%, followed by BS at 200 µg/mL, with 77.15%. BVE at 200 µg/mL (89.95%) acted similarly, with 200 µM 5-FU (88.63%) and 20 µM DOX (91.97%).

Table 1 and Figure 4 show that, in normal cells (HUVEC), both berberine and BVE have no cytotoxic effects (IC50 >> 400 µg/mL). Our results are similar to those from the scientific literature [18].

Figure 5 shows the influence of BVE and BS on cancer cell viability compared to standard oncolytic drugs in the same concentration range (12.5–200 µg/mL for BVE and BS, 12.5–200 µM for 5-FU and CisPt, and 1.25–20 µM for DOX) [17].

Generally, the tumor cell viability diminution was higher after 48 h than 24 h. In all cases, BS showed a higher cytotoxicity than BVE. In HT-29 and LoVo cells, the BS activity was lower than 5-FU after both exposure times, while in HEP G2, the BS activity was higher than 5-FU after 24 h and lower after 48 h (Figure 5A–F). The PE/CA-PJ49 cell viability decreased in the following order: be > CisPt > BS after 24 and 48 h of treatment (Figure 5K,L). In MDA-MB-231, the cell viability after 24 h decreased in the order of BVE > DOX > BS; after 48 h, BVE acted to a slightly higher extent than DOX (Figure 5G,H). After 24 h, the SK-OV-3 cell viability decreased in the order of DOX > BVE > CisPt > BS, while after 48 h, the previously mentioned order changed: DOX > BVE > BS > CisPt (Figure 5I,J). These effects are due to the potential synergism between the phytochemicals in BVE [19]. 

### 2.7. Statistical Analysis

The Pearson correlation shows that BVE24 is highly correlated with BS24, BVE48 and BS48 (r = 0.906, r = 0.910, r = 0.866, *p* < 0.05). BS24 exhibits a strong correlation with 5-FU24, BVE48 and BS48 (r = 0.813, r = 0.935, r = 0.928, *p* < 0.05). BVE48 has a considerable correlation with BS48 (r = 0.955) and a moderate one with 5-FU48 (r = 0.788), *p* < 0.05. Moreover, CisPt24 significantly correlates with DOX24, CisPt48 and DOX48 (r = 0.934, r = 0.997, r = 0.837, *p* < 0.05), CisPt48 with DOX48 (r = 0.830, *p* < 0.05), DOX24 with CisPt48 and DOX48 (r = 0.918, r = 0.920, *p* < 0.05), and 5-FU24 with BVE48 and BS48 (r = 0.866, r = 0.853, *p* < 0.05); 5FU24 also shows a moderate correlation with 5-FU48 (r = 0.788, *p* < 0.05). The place of each cytotoxic agent linked to the cell type is shown in Figure 6A, and the similarities between them are displayed in Figure 6B. 

The correlations between the bioactive phytoconstituents—total phenolic content (TPC) and total phenolic acid (TPA)—and their pharmacological potential are detailed in Appendix A. Their dual redox behavior could explain the antiproliferative effect on tumor cells, leading to decreasing viability; the prooxidant effect of phytochemicals is responsible for the BVE cytotoxicity. The antioxidant effect, measured by three methods, shows a substantial positive correlation with the TPC and TPA (r = 0.972–0.994, *p* < 0.05). Moreover, the variable parameters determined by all three methods (DPPH, FRAP, ABTS) are intercorrelated (r = 0.997–0.998, *p* < 0.05) and show a substantial negative correlation with the antiproliferative activity (r = −[0.951–0.999], *p* < 0.05). Similarly, the TPC and TPA display a significant negative correlation with the cell viability diminution (r = −[0.970–0.997], *p* < 0.05 (Appendix A). The outstanding capacity of *B. vulgaris* for scavenging ABTS, hydroxyl radicals, and DPPH is due to berberine and phenolic compounds with dual redox behavior that can act synergistically in the extract [2]. Recent studies showed that galangin and berberine in a synergic combination might induce esophageal carcinoma cells’ apoptosis through cell cycle arrest in the G2/M phase via oxidative stress [19]. Moreover, apigenin, gallic acid, and berberine have immunomodulatory potential and could be helpful as immune checkpoint inhibitors and fight cancers via multiple targets [20].

The accessed literature data regarding the cytotoxic effects of berberine evaluated in vitro in various tumor cells, potential mechanisms, and IC_50_ values are synthesized in Table 4.

In the present study, the BS IC_50_ against HEP G2 was slightly over 50 ug/mL, being around that registered in Table 4; the same was true for the colon carcinoma (LoVo), colon cancer (SK-OV-3) and tongue squamous cell carcinoma (PE/CA-PJ49) cell lines. Moreover, Table 4 indicates that the IC_50_ of BS against OSCC was 18–136 µM, and our value belongs to this range. For the MDA-MB-231 (breast cancer) cell line, the IC_50_ was > 25 µg/mL after 24 h and 12.5 after 24 h. Similar studies investigated the anticancer effects *of B. vulgaris* extract and berberine chloride in other cancer cell lines, evaluating the cell viability after 24, 48, and even 72 h and reporting various IC_50_ values [18,34,35]. In HEP G2 (liver cancer), Caco-2 (colon cancer), and MCF-7 (breast cancer), the IC_50_ values for barberry extract were 68.02 > 49.96 > 15.61 µg/mL, and for berberine chloride, lower values were recorded: 65.86 > 17.64 > 15.93 µg/mL [34]. After 48 h, the IC_50_ values drastically decreased: 5.55, 3.84, and 4.44 µg/mL for berberis extract vs. 11.49, 5.1, and 4.43 µg/mL for berberine chloride. Moreover, in HEP G2 and CaCo2, the antitumor activity of berberis extract was stronger than that of berberine chloride [34]. In our study, both BVE and BS had moderate cytotoxicity. Another research team analyzed the cytotoxicity of *B. vulgaris* extract in 70% ethanol on breast cancer cell lines (MCF-7) after 24, 48, and 72 h and obtained significantly higher IC50 values, respectively, 4000, 2000 and 1000 µg/mL [35]. Och et al. investigated the cytotoxic and proapoptotic properties of *B. thunbergii* extract and berberine on various hematopoietic cancer cell lines: acute promyelocytic leukemia (HL-60, HL-60/MX1, HL-60/MX2), myeloma (U266B1), acute lymphoblastic leukemia (CCRF/CEM and CEM/C1) and acute T cell leukemia (J45.01) [18]. After 24 h, the extract did not show cytotoxic effects in the tested cells, and the IC_50_ value of berberine was 80–250 µM [18]. However, tumor cells’ exposure to a high concentration of *B. thunbergii* extract influenced the activity of proapoptotic genes (upregulation of B2M, downregulation of BAD and BNIP2, and increased expression of BAX, BAK1, BIK, and CASP9i) in all the leukemia cell lines [18]. These phenomena suggest the potential detection of cellular apoptosis after an exposure longer than 24 h, and further experiments in the 72 and 96 h models are requested [18]. 

## 3. Materials and Methods

### 3.1. Materials

#### 3.1.1. Chemicals

All the chemicals were of analytical grade. Analytical standards of 31 compounds were purchased from Sigma-Aldrich, Schnelldorf, Germany. Methanol and ethyl alcohol, HPLC grade, were purchased from Merck, Bucharest, Romania; formic acid (98%) and ultrapure water (LC-MS grade) were also purchased from Merck (Merck Romania, Romania). The Pierce LTQ Velos ESI positive and negative ion calibration solutions (Thermo Fisher Scientific, Dreieich, Germany) calibrated the Orbitrap Mass Spectrometer.

The standard phenolic compounds (8 phenolic acids, 7 isoflavones, and 15 flavonoids), berberine sulfate hydrate, ethanol, sodium acetate, AlCl_3_, DPPH, ABTS ammonium salt, trichloroacetic acid, phosphate buffer (pH = 6.6), ascorbic acid, K_3_(FeCN)_6_ and FeCl_3_ were purchased from Sigma-Aldrich, Germany. Methanol and ethanol, potassium persulfate, formic acid (98%), and ultrapure water (LC-MS grade) were provided by Merck (Merck Romania SRL, Bucharest, Romania). The Pierce LTQ Velos ESI positive and negative ion calibration solutions (Thermo Fisher Scientific, Germany) calibrated the Orbitrap Mass Spectrometer.

In the in vitro studies in cell lines, various materials were used: Dulbecco’s Modified Eagle’s Medium (DMEM, PAN Biotech, Aidenbach, Germany), cell-washing medium HBSS (Hanks’ Balanced Buffer Solution), 200 mM L-glutamine, fetal bovine serum (FBS), 100 mM ethylenediaminetetraacetic acid (EDTA), phosphate-buffered saline (TFS), dimethyl sulfoxide (DMSO, Sigma-Aldrich, St. Louis, MO, USA), antibiotic mixture (10,000 U/mL penicillin and 10,000 µg/mL streptomycin) (Biochrom GmbH, Berlin, Germany), and Trypan Blue and CellTiter 96^®^ AQueous One Solution Cell Proliferation Assay (MTS) kit (Promega, Madison, WI, USA). The anticancer drugs (5-Fluorouracyl, Cisplatin, and Doxorubicin) and berberine sulfate hydrate were purchased from Sigma-Aldrich Chemie GmbH, Schnelldorf, Germany. Human adherent cancer cell lines of various histological origins were obtained from the international cell banks “European Collection of Authenticated Cell Cultures” (ECACC, Porton Down, Wiltshire, UK) or “American Type Culture Collection” (ATCC, Manassas, VA, USA) as follows: (a) MDA-MB-231 breast adenocarcinoma (ECACC, cat. no 92020424), (b) SK-OV-3 ovarian adenocarcinoma (ECACC, cat. no 91091004), (c) HEP G2 hepatocyte carcinoma (ECACC, cat. no 85011430), (d) PE/CA-PJ49 oral (tongue) squamous cell carcinoma (ECACC, cat. no 00060606), (e) HT-29 colon adenocarcinoma (ECACC, cat. no 91072201), and (f) LoVo colon adenocarcinoma (ATCC, cat. no CCL-229) [36]. As normal controls during the in vitro investigations, immortalized cells from human umbilical vein endothelial cells were used (adherent HUVEC cell line, kindly provided by Dr. Viviana Roman, Center of Immunology, “Stefan S. Nicolau” Institute of Virology, Bucharest, Romania). 

The *Daphnia magna* Straus for the in vivo studies originated from a culture maintained parthenogenetically at the Department of Pharmaceutical Botany and Cell Biology, Faculty of Pharmacy, “Carol Davila” University of Medicine and Pharmacy Bucharest, since 2012. 

#### 3.1.2. *B. vulgaris* Extract Preparation

*B. vulgaris* (L.) cortex was harvested in March 2023 from a local ecological crop in Oratia—Lat/Long (in decimal degrees): 45.445199, −27.013190—Buzau County, Romania. It was identified by Prof. Octavian Tudorel Olaru, Department of Pharmaceutical Botany and Cell Biology, and Prof. Cerasela Elena Gîrd, Department of Pharmacognosy, Phytochemistry and Phytotherapy, Faculty of Pharmacy, “Carol Davila” University of Medicine and Pharmacy, Bucharest. The voucher specimen is also preserved in the Department of Pharmacognosy, Phytochemistry, and Phytotherapy collection. Morphological peculiarities: the vegetable product is presented as flat or slightly recurved fragments; the inner face shows a bright yellow–green fluorescence in UV light (due to berberine). Organoleptic characteristics include a brown–gray color on the outside and a golden-yellow on the inside (due to berberine), which becomes brown through preservation (Appendix A), bitter taste, and no smell. As previously described, 50 g of powdered stem bark was subjected to reflux extraction with 50% ethanol (Sigma-Aldrich, Darmstadt, Germany) [37]. After filtration, the obtained extract (BVE) was concentrated in a rotary evaporator R100 with a vacuum pump V-700 (BUCHI Corporation, New Castle, DE, USA) and lyophilized (Christ Alpha 1-2/B Braun, BiotechInt, New Delhi, India). 

### 3.2. Total Polyphenol Content (TPC)

The Folin–Ciocalteu reagent was used following a spectrophotometric method described extensively in a previously published article [38]. The absorbances were measured at 725 nm (Jasco V-530 spectrophotometer, JASCO, Tokyo, Japan), and tannic acid was the standard for the calibration curve in a linear concentration range of 2–9 µg/mL. The TPC is expressed as mg Eq tannic acid/100 g BVE.

### 3.3. Total Phenolic Acid (TPA)

The quantification method was based on the phenolic acids that form nitro derivatives with nitrous acids. Our previously published article detailed it [37]. The absorbance was immediately measured at 525 nm (Jasco spectrophotometer, Japan) and compared to a sample that lacked the Arnow reagent. Chlorogenic acid (Sigma-Aldrich, Germany) was used as a standard for the calibration curve in the linear range of 11–53 μg/mL, with R^2^ = 0.9998. The total phenolic acid (TPA) content was expressed as mg chlorogenic acid equivalents per gram of extract (mg Eq chlorogenic acid/g BVE).

### 3.4. Identification and Quantification of Phenolic Constituents and Berberine

#### 3.4.1. Ultra-High-Performance Liquid Chromatography Coupled with High-Resolution Mass Spectrometry (UHPLC–HRMS/MS)

The phenolic profile of BVE was established based on non-targeted tandem mass spectrometry (MS-MS) using the hyphenated technique represented by Ultra-High-Performance Liquid Chromatography (UHPLC) coupled with the Q-Exactive High-Resolution Mass Spectrometer (HRMS). The same method was used to quantify selected phenolic compounds for each available analytical standard (Sigma-Aldrich, Germany). Our previously published study describes all the detailed data [17]. 

#### 3.4.2. High-Performance Liquid Chromatography

The separation was achieved on a reverse-phased analytical column (octadecylsilyl silica gel—C18, [25 × 0.4] mm i.d., 5 µm particle). The mobile phase consisted of a mixture of water and phosphoric acid = 0.1% (*v*/*v*) (solvent A) and 0.1% phosphoric acid in the acetonitrile (solvent B). The gradient used was as follows: 90%A/10%B, 0 min.; 90 → 78A/10 → 22/B, 0–15 min; 78 → 60A/28 → 40/B, 15–25 min.; 60 → 30%A/40 → 70%B, 15–40 min; 30 → 20%A/70 → 80%B, 40–55 min. The flow rate was 1.0 mL /min, associated with an injection volume of 20 μL and a monitoring wavelength of 330 nm.

### 3.5. Antioxidant Activity

#### 3.5.1. Diphenyl-1-Picrylhydrazyl Free Radical Scavenging Assay (DPPH)

Under an antioxidant, the purple free radical 2,2-diphenyl-1-picrylhydrazyl (DPPH) formed its corresponding yellow hydrazine. The absorbance value was measured at λ = 515 nm. The IC_50_ value was determined from the inhibition curves and their linear equations [39].

#### 3.5.2. Azinobis-3-Ethylbenzotiazoline-6-Sulfonic Acid Assay (ABTS)

The turquoise-colored ABTS radical resulted from a potent oxidizing agent (potassium persulfate) reaction with the ammonium salt of 2,2′-azino-bis(3-ethylbenzothiazoline-6-sulfonic acid). Under the action of the antioxidant, the intensity of the color was reduced to colorless. The absorbance was determined at λ = 734 nm, and the IC_50_ value was calculated from the inhibition curves and their linear equations [37].

#### 3.5.3. Ferric-Reducing Antioxidant Power Assay (FRAP)

The antioxidant analyte reacted with Fe^3+^, reducing to Fe^2+^, and imprinting blue. The coloration intensity was directly proportional to the antioxidant activity. The absorbance values were measured at λ = 700 nm (spectrophotometer Jasco V-530) and compared to the control (prepared under the same conditions without sample solution). It was expressed as an EC_50_ value; it represented the sample concentration at which the absorbance had a value of 0.5 or half the concentration at which the antioxidant activity was at a maximum, as determined by the trendline equation [40]. 

### 3.6. 48-h Acute Toxicity Test Using Daphnia Magna and Daphnia Pulex

The daphnids belonging to the species *Daphnia magna* and *Daphnia pulex* were chosen based on their size from parthenogenetic cultures maintained in an artificial medium for 24 h before testing [41,42]. The assay was performed in 24-well culture plates (Greiner Bio-One, Kremsmünster, Austria), with each well containing around 10 organisms. The samples were tested in six concentrations, ranging from 3.125 μg/mL to 100 μg/mL for BVE; as a positive control, BS was used from 0.625 to 20.0 μg/mL. The tests were duplicated, and lethality was assessed at 24 and 48 h. The 50% lethal concentrations (LC_50_) and the 95% confidence interval (CI95%) of the LC_50_ values were determined using GraphPad Prism v 5.1.2008 software (GraphPad Software, Boston, MA, USA) [17]. 

### 3.7. Daphnia Magna Embryonic Development Assay

The following concentrations were chosen for testing: BVE at 3.125 µg/mL and BS at 2.5 µg/mL, based on the results obtained in the viability test. The embryos were exposed to the sample solutions in the dark, maintaining a constant temperature and humidity of 25 °C and 75% RH, respectively. The experiments were carried out on culture plates with 48 wells (Greiner Bio-One, Kremsmünster, Austria). Every 24 h, the embryos were examined at a magnification of 80× under a microscope (bScope^®^ microscope, Euromex Microscope BV, Arnhem, The Netherlands) to assess the developmental stages and detect abnormalities compared to the untreated control [17].

### 3.8. In Vitro Anticancer Activity

#### 3.8.1. Cell Cultures and Treatments

The antiproliferative effect of the BVE hydro-ethanolic extract and BS standard was evaluated in vitro in six tumor cell lines (SK-OV-3, LoVo, HEP-G2, HT-29, MDA-MB-231, PE/CA-PJ49), with normal HUVEC cell, used as the control. All the cell lines were cultured in DMEM/F12 medium enriched with 2 mM L-glutamine and 10% fetal calf serum and antibiotics mixture (100 U/mL penicillin and 100 μg/mL streptomycin). They were incubated at 37 °C in a 5% CO_2_ humidified atmosphere. For the cytotoxicity assays, the cells were detached from the culture flasks and then cultivated in 96-well flat-bottom plates for 24 h until they reached around 70% confluence. Then, the cells were treated for various periods (24 h and 48 h) with different concentrations of BVE, BS, or oncolytic drugs (5-FU, CisPt, DOX) used as positive controls [43]. The BVE and BS stock solutions were prepared by dissolving them in a minimal amount of DMSO and preserved at 4 °C; all the working solutions were prepared from the stocks by serial dilutions with culture medium before each treatment assay [17].

#### 3.8.2. MTS Assay

The cytotoxic potential of BVE and BS was evaluated by a colorimetric cell viability method, the MTS assay, and it was assessed in both tumor and normal cells and compared with the action of oncolytic drugs: DOX, CisPt, and 5-FU [44].

All the assays were performed in triplicate using the CellTiter 96^®^ AQueous One Solution Cell Proliferation Assay (MTS) kit (Promega, USA). It contains a reagent mixture of two components: MTS [3-(4,5 dimethylthiazol 2 yl) 5 (3 carboxymethoxy phenyl) 2 (4 sulfophenyl) 2H tetrazolium] and PES (phenazine ethosulphate), a cationic dye with high chemical stability, which may be combined with MTS to form a stable solution [45]. The method’s principle is based on the ability of metabolically active cells to reduce MTS (a yellow tetrazolium salt) to the colored formazan, which is soluble in the culture medium and can be spectrophotometrically quantified at a 492 nm wavelength. Briefly, 1.5 × 10^4^ cells/well were cultured in 100 µL of the medium; after 24 h, the culture supernatants were discarded, and the cells were treated with increasing concentrations of BVE, BS, or reference drug solutions for 24 h or 48 h. At the expiration of the contact time, 20 µL of reagent mixture was added to each well, and the culture dishes were incubated for an additional 4 h at 37 °C, with gentle shaking every 20 min. Absorbance was read at λ = 492 nm with the Dynex ELISA reader (DYNEX Technologies—MRS, Chantilly, VA, USA) [46].

The cell viability was expressed as a percentage, was compared to the untreated cells (considered 100% viable), and was calculated according to the following formula:(1)$$\mathrm{Cell}\;\mathrm{viability}\;\%=100\;\times \;\frac{T-B}{U-B}$$
where *T* = optical density of treated cells, *B* = optical density of the blank (culture medium, in the absence of cells), and *U* = optical density of untreated cells.

The obtained results were expressed as the mean values from three different experiments (*n* = 3) ± standard deviation (SD) [47]. For the assessment of the DMSO cytotoxicity, the same experimental determinations were performed as in the MTS assay, and no impairment of cell viability was observed at concentrations lower than 1%.. Also, to observe the possible nonspecific reactions between BVE, BS, or drugs and MTS, their absorbance was determined without cells, and the values were extracted during the calculations.

### 3.9. Data Analysis

The statistically significant differences (at *α* < 0.05) between the various experimental groups were established by multiple pairwise comparisons using Dunn’s procedure from XLSTAT 2023.1.4. by Lumivero (Denver, CO, USA) [48].

The correlations between the bioactive constituents of the extracts and their antioxidant activity and cytotoxicity were determined using Principal Component Analysis [49] performed with XLSTAT 2023.1.4. by Lumivero (Denver, CO, USA) through Pearson correlation. A probability value *p* < 0.05 indicated a statistically significant difference [50]. 

## 4. Conclusions

This research investigated the autochthonous Berberidis cortex, obtaining a dry extract in 50% ethanol through successive reflux extraction, then solvent evaporation and freeze-drying. Through complex UHPLC–HRMS/MS and HPLC-DAD analysis of BVE, 40 phenolic constituents, including berberine, were identified. The main classes of phenolic metabolites (polyphenols and flavonoids) and bioactive representatives were also quantified. BVE’s significant antioxidant potential was revealed by in vitro evaluation of the radical scavenging ability and reducing power. Then, the acute toxicity tests highlighted BVE’s significant acute toxicity and teratogenicity in *Daphnia* sp. It also displayed moderate antiproliferative activity in various tumor cell lines and did not affect normal cells. Compared to BVE, berberine showed higher toxicity. It is essential to show that berberine sulfate reduced the viability in several tumor cell lines more than the standard anticancer drugs used as positive controls.

Strong and statistically significant correlations were recorded between the exposure time, concentration, phenolic metabolites content, antioxidant activity, and cytotoxicity of *B. vulgaris* stem bark dry hydro-ethanolic extract.

Our results could enrich the scientific database regarding the composition and pharmacological properties of autochthonous Berberidis cortex. Further research could explore the acute toxicity and teratogenicity of BVE and berberine using other animal models and investigate their anticancer activity mechanisms in various other tumor cell lines. 

## Figures and Tables

**Figure 1 molecules-29-02053-f001:**
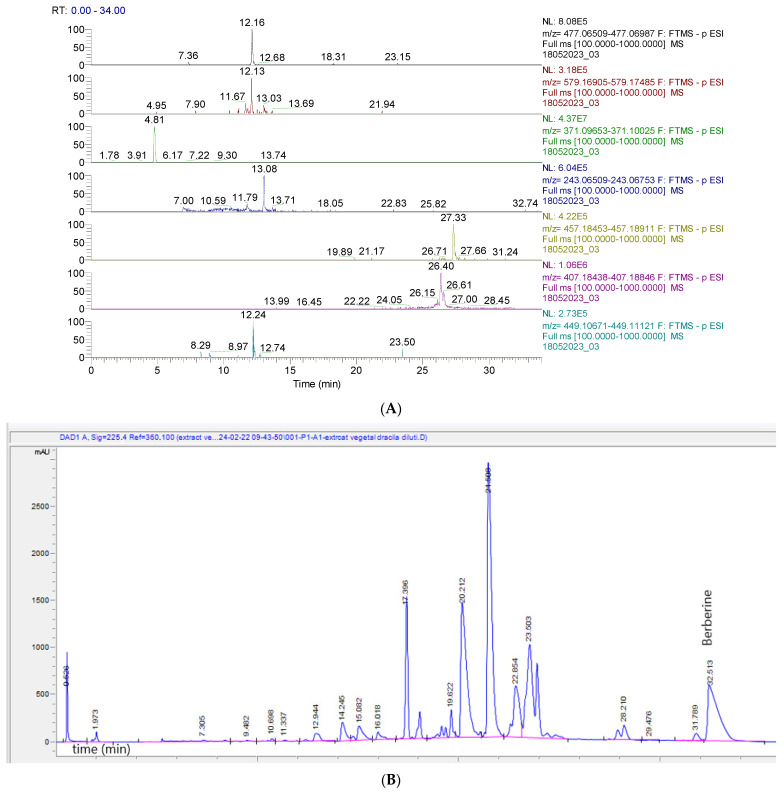
(**A**). UHPLC–HRMS/MS chromatogram of phytochemicals identified in BVE; from top to bottom: quercetin 3-O-glucuronide (*m*/*z* = 477.06749, RT = 12.16), narirutin (naringenin-7-O-rutinoside) (*m/z* = 579.17195, RT = 12.13), hydroxyferulic acid (*m/z* = 371.09839, RT = 4.81), piceatannol (*m/z* = 243.06631, Rt = 13.08), lignan (*m/z* = 457.18682, RT = 27.33), lehmannin (*m/z* = 407.18642, RT = 26.40), taxifolin 3-O-rhamnoside (*m/z* = 449.10896, RT = 12.24). (**B**)**.** HPLC-DAD chromatogram of BVE; berberine has an RT = 32.513.

**Figure 2 molecules-29-02053-f002:**
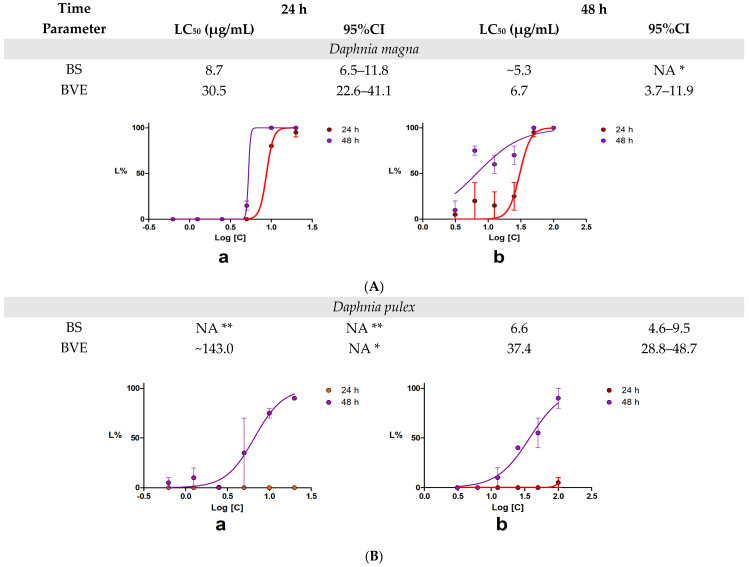
The results of the 48 h acute toxicity test using *Daphnia* sp. Lethality curves obtained after 48 h exposure of *Daphnia* sp. to BS (a) and BVE (b): *D. magna* (**A**) and *D. pulex* (**B**). BS—berberine sulfate hydrate; BVE—*B. vulgaris* stem bark dry hydro-ethanolic extract; NA—the values could not be calculated: * The interval is vast. ** The values could not be calculated as the maximum L% was 10%.

**Figure 3 molecules-29-02053-f003:**
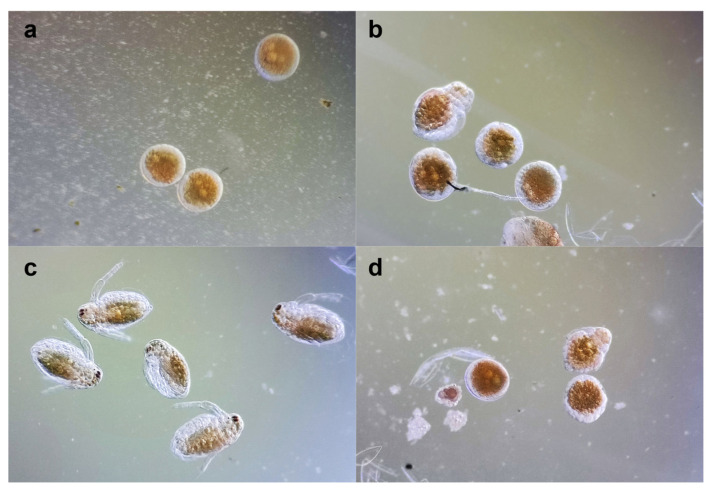
*Daphnia magna* embryonic development assay: (**a**) embryos before testing; (**b**) embryo development after 24 h in BS 2.5 μg/mL; (**c**) embryo development after 48 h in BS 2.5 μg/mL; (**d**) embryo development after 48 h in BVE 3.125 μg/mL.

**Figure 4 molecules-29-02053-f004:**
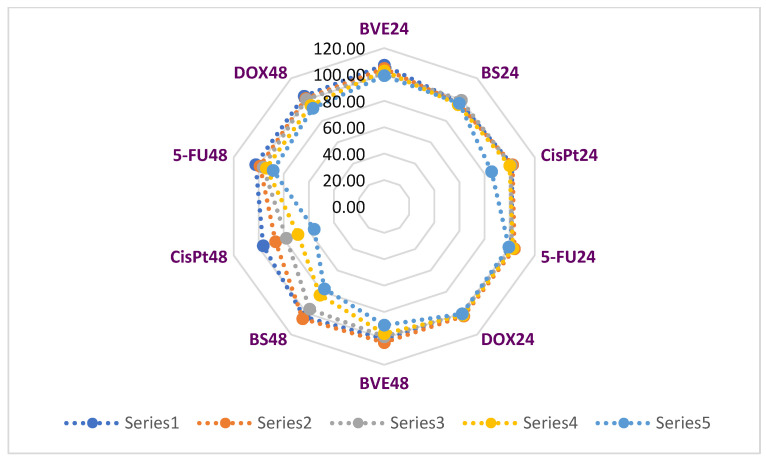
The effects of BVE and BS on normal endothelial cell (HUVEC) viability (%) compared to oncolytic drugs after 24 and 48 h. Series 1–5 = concentration range: 12.5–200 µg/mL for BVE and BS, 12.5–200 µM for 5-FU and CisPt, and 1.25–20 µM for DOX. BVE—dry hydro-ethanolic extract of *Berberis vulgaris* stem bark; BS—berberine sulfate hydrate; CisPt—Cisplatin; DOX—Doxorubicin, 5-FU—5-Fluorouracil; 24 and 48—treatment time (24 and 48 h).

**Figure 5 molecules-29-02053-f005:**
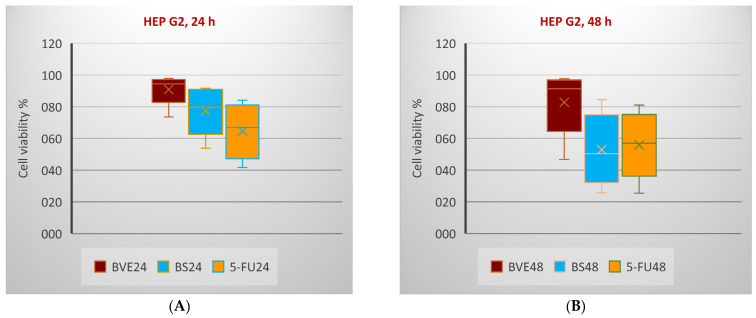
Box and Whisker plots displaying tumor cell viability % (F1 axis) after 24 h (**A**,**C**,**E**,**G**,**I**,**K**) and 48 h (**B**,**D**,**F**,**H**,**J**,**L**) following treatments with BVE, BS, and standard anticancer drugs: (**A**,**B**): HEP G2; (**C**,**D**): HT-29; (**E**,**F**): LoVo; (**G**,**H**): MDA-MB-231; (**I**,**J**): SK-OV-3; (**K**,**L**): PE/CA-PJ49. HEP G2—human hepatocellular carcinoma; HT-29 and LoVo—human colon adenocarcinomas; MDA-MB-231—human breast adenocarcinoma; PE/CA-PJ49—human squamous tongue carcinoma; SK-OV-3—human ovary adenocarcinoma; BVE—dry hydro-ethanolic extract of *Berberis vulgaris* stem bark; BS—berberine sulfate hydrate; CisPt—Cisplatin; DOX—Doxorubicin, 5-FU—5-Fluorouracil; 24 and 48—treatment time (24 and 48 h).

**Figure 6 molecules-29-02053-f006:**
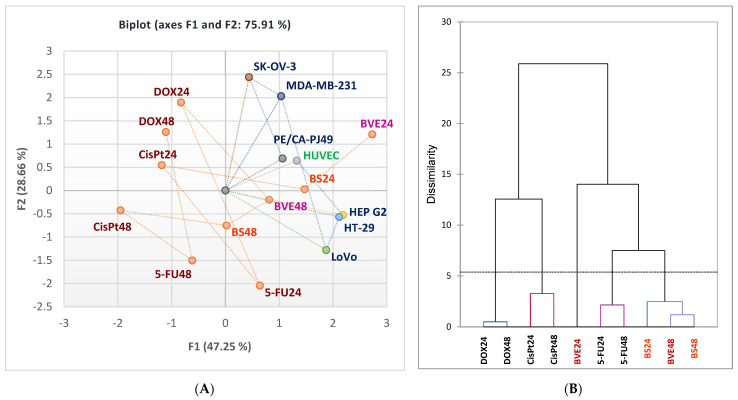
(**A**). Symmetric biplot displaying the correlation between the cytotoxic effects of BVE, BS, and anticancer drugs on normal and tumor cells after 24 and 48 hr of treatment. (**B**). AHC-Dendrogram. HEP G2—human hepatocellular carcinoma; HT-29 and LoVo—human colon adenocarcinomas; MDA-MB-231—human breast adenocarcinoma; PE/CA-PJ49—human squamous tongue carcinoma; SK-OV-3—human ovary adenocarcinoma; BVE—*B. vulgaris* dry extract; BS —berberine sulfate hydrate; CisPt—Cisplatin; DOX—Doxorubicin, 5-FU—5-Fluorouracil; 24 and 48—treatment time (24 and 48 h).

**Table 1 molecules-29-02053-t001:** Total polyphenols content, total phenolic acids, and antioxidant activity of BVE.

**Phenolic Compounds**			
**Total Polyphenols** **(mg Eq Tannic Acid/100 g Extract)**		**Total Phenolic Acids** **(mg Eq Chlorogenic Acid/100 g Extract)**	
17.6780 ± 3.9320		3.3886 ± 0.3481	
**Antioxidant Activity**			
**IC_50_DPPH (mg/mL)**	**IC_50_ABTS (mg/mL)**		**EC_50_FRAP (mg/mL)**
0.2610	0.0442		0.1398

DPPH—2,2-diphenyl-1-picryl-hydrazine; ABTS—2,20-azinobis-3-ethylbenzotiazoline-6-sulfonic acid; FRAP—ferric reducing antioxidant power.

**Table 2 molecules-29-02053-t002:** The phytochemicals identified in *B. vulgaris* stem bark dry extract (BVE) by UHPLC–HRMS/MS and HPLC-DAD.

Nr. Crt.	Identified Compound	Phytochemical Classification	Chemical Formula	Adduct Ion/ Monitored Negative Ion (*m/z*)	Retention Time (min)	Content (µg/g)
**1**	Quercetin	Flavonoid	C_15_H_10_O_7_	301.0354	15.01	28.42
**2**	Rutin (quercetin 3-O-rutinoside)	Flavonoid	C_27_H_30_O_16_	609.14613	12.39	72.41
**3**	Apigenin	Flavonoid	C_15_H_10_O_5_	269.04502	16.71	10.45
**4**	Kaempferol	Flavanol	C_15_H_10_O_6_	285.04049	16.51	68.74
**5**	6-Methoxyluteolin (Nepetin)	Flavonoid	C_16_H_12_O_7_	315.05105	16.75	-
**6**	Naringenin	Flavanone	C_15_H_12_O_5_	271.06122	15.46	90.41
**7**	Hesperitin	Flavonoid	C_16_H_14_O_6_	301.07179	13.71	44.00
**8**	Galangin	Flavonoid	C_15_H_10_O_5_	269.04557	16.71	67.21
**9**	Genistein	Isoflavone	C_15_H_10_O_5_	269.04502	16.73	-
**10**	Glycitein	Isoflavone	C_16_H_12_O_5_	283.06122	11.15	19.21
**11**	Gallic acid	Hydroxybenzoic acid	C_7_H_6_O_5_	169.01427	1.70	540.00
**12**	Chlorogenic acid/Neochlorogenic	Cinnamate ester	C_16_H_18_O_9_	353.08783	6.08	10.54
**13**	Ferulic acid	Hydroxycinnamic acid	C_10_H_10_O_4_	193.05066	9.94	39.36
**14**	AbsCisPtic acid	Terpenoid	C_15_H_20_O_4_	263.12891	14.76	8.61
**15**	*p*-Coumaric acid	Hydroxycinnamic acid	C_9_H_8_O_3_	163.03954	8.80	30.33
**16**	Syringic acid	Hydroxybenzoic acid	C_9_H_10_O_5_	197.04555	8.73	3.35
**17**	Afrormosin	Isoflavone	C_17_H_14_O_5_	297.07687	17.17	-
**18**	Kaempferol-3-O-rutinoside	Flavonol glycoside	C_27_H_30_O_15_	593.15122	9.35	-
**19**	Kaempferol (luteolin)-O-glucoside/ isomers	Flavonoid	C_21_H_20_O_11_	447.09331	13.56	-
**20**	Vitexin (apigenin 8-C-glucoside)/isovitexin	Flavonol glycoside	C_21_H_20_O_10_	431.09839	11.98	-
**21**	Azelaic acid	Dicarboxylic acid	C_9_H_16_O_4_	187.09761	13.99	-
**22**	Apigenin 7-O-glucosylglucoside	Flavonoid	C_27_H_30_O_15_	593.15122	9.45	-
**23**	Rosmarinic acid	Ester of caffeic acid	C_18_H_16_O_8_	359.07727	13.42	-
**24**	Carnasol	Diterpene	C_20_H_26_O_4_	329.17586	18.83	-
**25**	Rosmadial/Isomeri	Diterpene lactone	C_20_H_24_O_5_	343.15510	20.38	-
**26**	Rosmanol methyl ether	Diterpene	C_21_H_28_O_5_	359.18640	22.19	-
**27**	Quercetin 3-O-glucuronide	Flavonol glucuronide	C_21_H_18_O_13_	477.06749	12.16	23.04
**28**	Narirutin (naringenin-7-O-rutinoside)	Flavonol glycoside	C_27_H_32_O_14_	579.17195	12.13	-
**29**	Apigenin-7-O-glucuronide	Flavonoid-7-O-glucuronides	C_21_H_18_O_11_	445.07763	13.29	-
**30**	Procyanidine B1/B2	Flavonoid	C_30_H_26_O_12_	577.13515	16.24	-
**31**	Sinapic acid	Hydroxycinnamic acid	C_11_H_12_O_5_	223.06122	10.33	-
**32**	Hidroxyferulic acid/Isomers	Hydroxycinnamic acid	C_16_H_20_O_10_	371.09839	4.81	-
**33**	Valerenic acid	Sesquiterpenoid	C_15_H_22_O_2_	233.15473	21.33	-
**34**	Lehmannin	Flavanone	C_25_H_28_O_5_	407.18642	26.40	-
**35**	Ginkgetin	Flavone	C_32_H_22_O_10_	565.11404	7.25	-
**36**	Taxifolin 3-O-rhamnoside	Flavonoid	C_21_H_22_O_11_	449.10896	12.24	-
**37**	Piceatannol	Stilbenoid	C_14_H_12_O_4_	243.06631	13.08	-
**38**	Lignan	Polyphenolic compound	C_25_H_30_O_8_	457.18682	27.33	-
**39**	Cyanidin 3-O-arabinoside	Anthocyanidin-3-O-glycoside	C_20_H_19_ClO_10_	453.05942	7.17	-
**40**	Berberine	Isoquinoline alkaloid	C_20_H_18_NO_4_	-	32.51	78.95

**Table 3 molecules-29-02053-t003:** The cytotoxicity of BVE and BS in normal cell and tumor cell lines (expressed as cell viability %) after 24 and 48 h of exposure.

Concentration (µg/mL)	24 h						48 h					
	BVE			BS			BVE			BS		
	V (%)	SD	IC_50_ (µg/mL)	V (%)	SD	IC_50_ (µg/mL)	V (%)	SD	IC_50_ (µg/mL)	V (%)	SD	IC_50_ (µg/mL)
HUVEC												
6.25	109.38	5.13		100.60	4.15		104.59	5.04		107.52	5.67	
12.5	107.10	5.38		97.10	5.91		100.54	4.35		103.36	6.91	
25	104.60	1.06		96.28	4.58		103.16	3.56		105.17	3.38	
50	101.16	0.09	>>400	99.53	4.33	>>400	98.71	5.03	>>400	96.18	2.98	>400
100	102.78	3.54		95.35	2.21		96.38	3.59		82.99	6.04	
200	99.10	7.94		96.72	1.86		89.95	4.76		77.15	4.99	
400	86.53	4.69		84.60	2.30		75.48	0.09		56.10	0.09	
HEP G2												
6.25	99.57	6.47		97.46	6.76		98.31	0.33		96.91	5.81	
12.5	97.94	4.25		91.71	8.04		97.60	2.06		84.48	0.54	
25	96.43	0.12		90.43	4.08		96.03	6.67		65.45	2.39	
50	94.49	1.98	>400	79.68	5.30	>200	91.33	1.90	>100	50.48	2.28	>50
100	91.83	2.97		71.19	7.51		81.87	3.20		38.32	0.22	
200	73.62	0.52		53.77	5.48		46.76	1.95		25.69	2.44	
400	52.86	0.93		42.16	2.80		22.62	3.36		21.82	6.02	
HT-29												
6.25	100.78	2.80		100.08	1.27		99.90	4.08		98.23	5.95	
12.5	98.86	7.60		98.63	2.90		97.39	0.00		95.90	2.72	
25	98.62	4.80		94.72	3.54		94.81	1.53		92.29	1.93	
50	97.18	3.59	>>400	83.49	4.75	>400	92.31	3.34	>>400	86.48	5.16	>400
100	95.08	1.16		70.77	0.60		90.62	1.25		72.18	2.44	
200	91.06	0.69		66.59	4.75		86.62	0.28		61.43	2.21	
400	85.04	3.69		60.80	6.76		77.31	5.45		54.02	6.56	
LoVo												
6.25	98.74	4.77		97.08	5.37		92.46	3.67		95.64	4.80	
12.5	93.91	0.74		91.98	3.52		89.06	6.32		91.86	2.73	
25	90.85	7.54		88.50	4.96		84.68	8.47		82.13	6.31	
50	85.13	4.94	>200	79.91	0.87	>50	72.68	4.55	>50	65.02	5.69	>50
100	75.97	2.35		38.60	1.85		48.39	0.63		28.19	1.39	
200	53.72	4.45		26.22	2.10		14.72	0.00		15.77	1.14	
400	25.05	4.57		15.57	8.53		3.09	2.66		8.08	2.91	
MDA-MB-231												
6.25	96.83	4.41		87.73	6.43		75.86	4.48		71.40	3.52	
12.5	90.33	2.57		76.14	0.55		73.74	3.67		62.02	0.95	
25	87.71	0.37		66.43	5.51		70.81	0.15		45.93	2.57	
50	82.02	2.20	>100	49.76	1.10	>25	60.88	1.84	>50	37.83	6.68	>12.5
100	69.16	2.94		23.19	2.02		45.27	3.97		20.29	2.13	
200	53.71	3.67		20.67	0.37		25.65	1.47		14.39	2.06	
400	17.21	3.49		10.43	4.04		2.99	0.44		8.54	0.81	
PE/CA-PJ49												
6.25	99.85	6.60		97.37	4.32		96.00	2.18		91.19	5.00	
12.5	93.17	6.89		91.01	8.44		87.26	0.51		77.49	6.21	
25	86.25	7.68		78.14	5.90		78.68	7.12		62.58	0.51	
50	80.89	2.54	>100	63.65	2.81	>50	69.09	5.95	>50	50.24	4.70	>50
100	61.92	1.17		42.84	0.34		49.39	5.35		35.37	1.91	
200	43.51	2.40		23.86	0.34		23.20	2.06		12.96	0.88	
400	12.25	2.54		9.75	1.71		4.70	0.51		3.95	0.66	
SK-OV-3												
6.25	99.72	4.94		99.03	4.61		93.80	6.76		91.34	5.80	
12.5	92.28	5.89		90.99	4.63		90.86	7.09		82.43	5.30	
25	89.38	6.28		83.36	5.25		83.56	0.55		78.78	6.68	
50	83.35	5.84	>400	79.62	6.04	>200	73.17	2.56	>100	67.19	7.40	>50
100	78.30	2.95		66.62	1.87		58.24	3.52		33.37	6.76	
200	72.10	7.07		57.21	4.81		30.87	3.38		26.46	2.47	
400	55.83	3.83		39.90	1.25		16.29	0.37		10.05	0.80	

BVE = *B. vulgaris* stem bark dry hydro-ethanolic extract, 24 and 48 h = cell line exposure time (hours) to the different BVE concentrations (µg/mL). HUVEC—human umbilical endothelial cell; HEP G2 —human hepatocellular carcinoma; HT-29 and LoVo—human colon adenocarcinomas; MDA-MB-231—human breast adenocarcinoma; PE/CA-PJ49—human squamous tongue carcinoma; SK-OV-3 —human ovary adenocarcinoma; SD—standard deviation. Interpretation of the IC_50_ values is based on that of the National Cancer Institute [16]: IC_50_ ≤ 20 μg/mL—strong cytotoxic properties, IC_50_ = 21–200 μg/mL—moderate cytotoxicity, IC_50_ = 201–500 μg/mL—low cytotoxicity and IC_50_ ≥ 500 μg/mL—no cytotoxic activity. Data are expressed as the mean values ± standard deviations (SD) of three experiments (*n* = 3).

**Table 4 molecules-29-02053-t004:** In vitro cytotoxicity of berberine in various tumor cell lines, based on the literature data.

Cancer Cell Line	Cytotoxic Responses	Berberine Concentration	IC_50_ Value	Reference
Liver cancer HEP G2 SMMC-7721 Bel-7402	- Decreases the cell viability in a time- and dose-dependent manner.	3.125, 6.25, 12.5, 25, 50 and 100 µM	HEP G2—34.5 µM, SMMC-7721—25.2 µM Bel-7402—53.6 µM	[21]
Ehrlich ascites carcinoma EAC	- Increases apoptotic cells (at 10 µg/mL); - Inhibits DNA synthesis; - Changes the morphology of dsDNA; - Induces cell death (at 50 and 100 µg/mL).	10, 50 and 100 µg/mL	<1 µg/mL	[22]
Dalton’s lymphoma ascites DLA	- Induces cytotoxicity of 44% at a concentration of 1 mg/mL; - At lower concentrations, it caused dose-dependent cytotoxicity in DLA cells.	100–1000 mg/mL	NA	[23]
Breast cancer MCF-7 MDA-MB-231	Dose- and time-dependent inhibitory effects of cancer cell proliferation: - Increases apoptotic ratio; - Stimulates caspase-3 activity and alteration in cell morphology; - Increases ROS generation; - Induces overexpression of p53.	10–100 µM 10–100 µg/mL	NA MCF7—15.93 ug/mL	[24]
Ovarian cancer CsSki, SiHa, HeLa	- Inhibits the invasion of CsSki, HeLa, and SiHa cells in a dose-dependent manner; - Inhibits the migration of CsSki, SiHa, and HeLa cells; - Decreases the SiHa cell motility.	20 µM	NA	[25]
Prostate cancer LNCaP PC-82	- Decreases the cell viability and induces programmed necrosis and apoptosis in a dose-dependent manner.	1–100 µM	NA	[26]
Rat glioma C6	Cytotoxic effects occur in a time- and dose-dependent manner, as follows: - Alters the cell morphology; - Promotes the caspase-3, -8, and -9 activity; - Increases the production of ROS; - Induces apoptotic cell death.	100 µM	NA	[27]
Colorectal carcinoma HCT116, SW480 LoVo	- In a concentration- and time-dependent manner, the cancer cell growth was inhibited via programmed death.	0–100 µM for 24–72 h	NA	[28]
Human prostate cancer LNCaP, PC-3	- Blocks growth and proliferation in cancer cells in a time- and concentration-dependent manner; - Induces apoptotic cell death.	0, 5, 10, 20, 50, and 100 µM	LNCap cells: 60 µM PC-3 cells: ≥100 µM	[26,29]
Lung cancer A549	- Did not show a cytotoxic effect on the A549 cells (up to 24 h); - Slight cytotoxicity was observed after 48 h of exposure (at 20 and 40 µM).	2.5–40 µM	NA	[30]
Human esophageal cancer YES-2	- Reduces cell viability and proliferation; - Inhibits production of interleukin 6; - All effects are dose- and time-dependent.	8–32 µM	NA	[31]
Oral cancer: OC2 KB	- Inhibits activator protein 1; - Exhibits anti-inflammatory effects by reducing the production of cyclooxygenase-2 (COX-2) and prostaglandin E2 (PGE2).	1, 10, and 100 µM (2–12 h)	NA	[32]
Human OSCC: HSC-2, HSC-3, HSC-4 Human Promyelocytic Leukemia: HL-60	- Increases apoptotic cells; - Induces DNA fragmentation; - Stimulates caspase-3, -8 and -9 and proapoptotic BAD protein; - In HSC-2 cells, BAD protein increase was not available.	10, 20 and 80 µM	18–136 µM	[32]
Mouse melanoma K1735-M2	- Inhibitory effect on cell proliferation is dose- and time-dependent; - 50% of growth inhibition was observed after 72 and 96 h of exposure.	0, 10, 25, 50, 75, and 100 µM	NA	[33]

NA—Not available.

## Data Availability

Data are available in the manuscript and Appendix A.

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
