# Peer review of "Antioxidant and Cytotoxic Properties of Berberis vulgaris (L.) Stem Bark Dry Extract"

_molecules, 2024, doi:10.3390/molecules29092053_

Round 1
Reviewer 1 Report
Comments and Suggestions for Authors
Author Response
Dear Reviewer 1, Thank you so much for your time and accurate review report, with valuable comments to improve the quality of our MS.
We rectified our MS according to your comments, point by point, marking all changes with track changes and indicating our response to them.
Please find it in the attachment.

Reviewer 2 Report
Comments and Suggestions for Authors
Comments to the Author
The authors have investigated the “Antioxidant and Cytotoxic Properties of Berberis vulgaris (L.) Stem Bark Dry Extract”. I recommend reconsidering after the minor revision.
The main comments are as follows:
Materials and Methods-
1. Line 99, 110-112. Please check the order of Table 1 and Table 2.
Results and discussion-
1. Line 99, 315-317. Please add references for this part.
2. Line 369. Generally, most plants extract active substances by using 80% ethanol as solvent. In this part, the obtained extract (BVE) was extracted with 50% ethanol. Why 50% ethanol was used as extractant, which has higher extraction efficiency or higher extract purity?
3. Line 385 and Line 391. The total Polyphenols Content (TPC) and total Phenolic Acids (TPA) were determined, the calibration curve in a linear concentration range of 2–9 µg/mL and 11-53 µg/mL, please supplement the standard curve equation for TPC and TPA determination if it is convenient.
4. Line 408. “The mobile phase consisted of a mixture of water and phosphoric acid = 0.1% (v/v) (solvent A) and 0.1% phosphoric acid in the acetonitrile (solvent B).” What is the reason for adding 0.1% phosphoric acid to mobile phases A and B?
Supplementary Material -
Table S3. What does PAC mean in the table? Please check “PAC” in this table, is PAC or TPA?
Author Response
Dear Reviewer 2, Thank you so much for your time and accurate review report, with valuable comments to improve the quality of our MS.
We rectified our MS according to your comments, point by point, marking all changes with track changes and indicating our response to them.
Please find it in the attachment.
